# Effect of Silicon–Manganese Deoxidation on Oxygen Content and Inclusions in Molten Steel

**Tianle Song** [1]**, Zhongliang Wang** [1]**, Yanping Bao** [1,*]**, Chao Gu** [1,*] **and Zefeng Zhang** [2]

1   State Key Lab of Advanced Metallurgy, University of Science & Technology Beijing, Beijing 100083, China
2   HBIS Materials Technology Research Institute, Gaocheng District, Shijiazhuang 052165, China
*   Correspondence: baoyp@ustb.edu.cn (Y.B.); guchao@ustb.edu.cn (C.G.)

**Abstract:** In order to improve the cleanliness of steel, non-aluminum deoxidation processes have begun to replace aluminum deoxidation processes. Although the aluminum deoxidation process can reduce the oxygen content in steel to $<10 \times 10^{-6}$, this deoxidation method causes fatigue failure resulting from the formation of large-grained spherical (Ds-type) inclusions composed of calcium–aluminate. It also tends to lead to nozzle blockage during casting. Given the above problems, this study conducted an in-depth investigation of silicon–manganese deoxidation. Thermal experiments and thermodynamic calculations were used to assess the impact of different Mn–Si ratios on the oxygen content and inclusion characteristics during the deoxidation process of molten steel with different initial oxygen contents. The experimental samples were analyzed using an oxygen–nitrogen–hydrogen analyzer, a direct reading spectrometer, and an automatic scanning electron microscope. After that, the samples were electrolyzed to observe the 2D morphology and 3D morphology of the inclusions using scanning electron microscopy. Finally, thermodynamic calculations were carried out using FactSage to verify the experimental results. The results indicated that, regardless of the initial oxygen content, silicon–manganese deoxidation maintained the total oxygen content at $35 \times 10^{-6}$. It effectively managed the plasticization of inclusions in molten steel, predominantly yielding spherical silicates while minimizing Al-containing inclusions. Nevertheless, as the initial content of [O] increased, the size and density of the silicate inclusions in the steel also increased. An optimal point in the number and size of inclusions was observed with an increased Mn–Si ratio. Moreover, the combined utilization of silicon–manganese deoxidation, diffusion deoxidation, and vacuum deoxidation enabled ultra-low oxygen content control of molten steel.

**Keywords:** silicon–manganese deoxidation; inclusion; cleanliness; manganese–silicon ratio



## 1. Introduction

In recent years, the manufacturing industry has witnessed rapid growth, resulting in an increased demand for high-quality steel [1–3]. To meet the stringent performance requirements of high-quality steel, controlling harmful non-metallic inclusions formed during the steelmaking process is essential. In alloy structural steel, brittle inclusions are the most harmful. The presence of brittle inclusions can lead to product failure due to cracking [4–8]. To mitigate the harmful effects of inclusions on steel quality, some scholars [8–13] have proposed two methods: controlling the total oxygen content in molten steel to reduce the number of inclusions or optimizing steel properties by improving the characteristics of inclusions in steel. Some scholars [14] have shown that there is excellent correspondence between the total oxygen content in steel and the number of inclusions. And the amount of inclusions affects the fatigue life of bearing steel. The fatigue life of bearing steel can be increased 100 times when the total oxygen content is reduced from 30 ppm to 5 ppm [15]. Aluminum can reduce the oxygen content of steel to a very low level. However, further reductions in the total oxygen content of ultra-high-purity bearing steel did not significantly improve its fatigue life and caused some inclusions generated

by aluminum deoxidation, such as $Al_2O_3$, magnesium–aluminum spinel, and calcium–aluminate, which are non-deformable in the conventional thermal processing temperature range of steel and are extremely harmful to steel. Non-deformable spherical alumina inclusions and calcium aluminate inclusions tend to be larger and have a substantial impact on the fatigue life of steel, potentially leading to premature failure [16–18]. Additionally, during the continuous casting of aluminum-deoxidized steel, particularly in small cross-sections, nozzle blockages and nodules frequently occur, resulting in thinner molten steel or even cut-off values [19–21]. Nodule composition analysis revealed that the chemical composition of nozzle-blocked powder closely resembled that of the inclusions in the steel, primarily consisting of $Al_2O_3$. However, adopting the silicon–manganese (Si–Mn) deoxidation process significantly improves the fluidity of molten steel and improves the phenomenon of nozzle clogging [22,23].

Reducing the oxygen content in steel is not the only way to improve the quality and cleanliness of steel. To further improve the quality of steel, silicon and manganese are used together for deoxidation. In the history of steelmaking development, the use of elements such as silicon [Si] and manganese [Mn] for deoxidation was among the earliest practices, but it often led to incomplete deoxidation. Presently, most reports have focused on [Si] and [Mn] deoxidation in actual production, with limited systematic and in-depth discussions on different molten steel conditions. Generally, it is believed that Si–Mn-deoxidized molten steel can effectively prevent the occurrence of calcium–aluminate inclusions while reducing the oxygen content of molten steel. Nonetheless, it is necessary to strictly control the number and size of silicate inclusions to ensure optimal steel performance [24–26]. Research by Wang et al. showed that under the same total oxygen content, the fatigue performance of bearing steel deoxidized by Si–Mn can match or even slightly surpass that of Al-deoxidized bearing steel [15]. Moreover, a study conducted by Gu et al. showed that, compared with steel deoxidized with Al, the silicate and the calcium aluminate size difference was larger and the silicate size of the steel deoxidized by Si–Mn was significantly smaller than the inclusions formed by deoxidation with Al [27].

In the current field of bearing steel production, calcium–aluminate inclusions generated by aluminum deoxidation have become a major source of fatigue failure. There is an urgent need to develop a non-aluminum deoxidation process instead of the traditional aluminum deoxidation process to improve the quality stability of bearing steel. This study primarily offered an in-depth investigation into the Si–Mn deoxidation of molten steel and quantitatively analyzed the effects of different Mn–Si ratios on the [O] content and inclusion characteristics during the deoxidation process in molten steel with varying initial [O] contents, utilizing thermal experiments and thermodynamic calculations.

## 2. Experimental Method

### 2.1. Experimental Design

To decrease the impact of Al in the experiment, different oxygen contents of molten steel were prepared using high-purity electrolytic iron and a certain amount of $Fe_2O_3$. The experiments were carried out in a 2 kg vacuum induction furnace equipped with argon-filling, alloying, water-cooling, and casting systems (Figure 1a). A vacuum induction furnace inductively heats the melt by means of a coil and the melt temperature is controlled by a voltage control system. In addition, different amounts of $Fe_2O_3$ and 700 g of electrolytic iron (Table 1) were placed in a magnesium crucible (inner diameter 48 mm, outer diameter 63 mm, height 135 mm) prior to the experiment. The crucible was placed inside an insulated graphite sleeve. Temperature measurements were taken using an infrared thermometer gun and the voltage was adjusted to keep the temperature stable during the process (Figure 2). Before heating the electrolytic iron, a mechanical pump evacuated the air pressure in the chamber to 10 Pa. Then, protective argon gas was injected into the chamber to bring it up to atmospheric pressure. The molten iron was obtained when the electrolytic iron reached 1650 °C. At this time, sample 1 was taken using a vacuum sampler. Low-carbon ferromanganese and low-carbon ferrosilicon were added to the molten steel together

through the feeding bin for deoxidation. Sample 2 was taken after the alloy was added for 1 min, sample 3 after 5 min, sample 4 after 15 min, and sample 5 after 30 min (Figure 2).

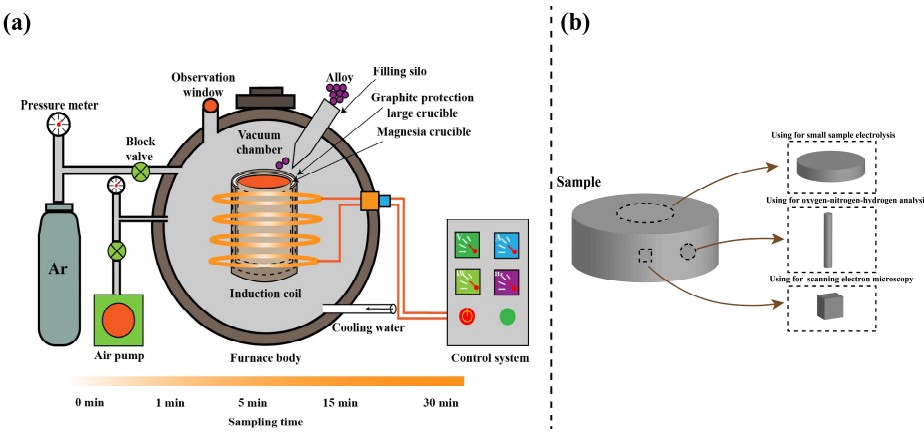

**Figure 1.** (**a**) Schematic illustration of 2 kg vacuum induction furnace and (**b**) sample research.

**Table 1.** Experimental materials' compositions, wt %.

| Sample | [C] | [Si] | [Mn] | [Al] | [O] |
|---|---|---|---|---|---|
| Electrolytic iron | 0.0005 | 0.0008 | 0.0003 | 0.0042 | 0.0130 |
| Low-carbon ferrosilicon | 0.01 | 76.92 | - | - | 0.0125 |
| Low-carbon ferromanganese | 0.65 | - | 80.8 | - | 0.0093 |

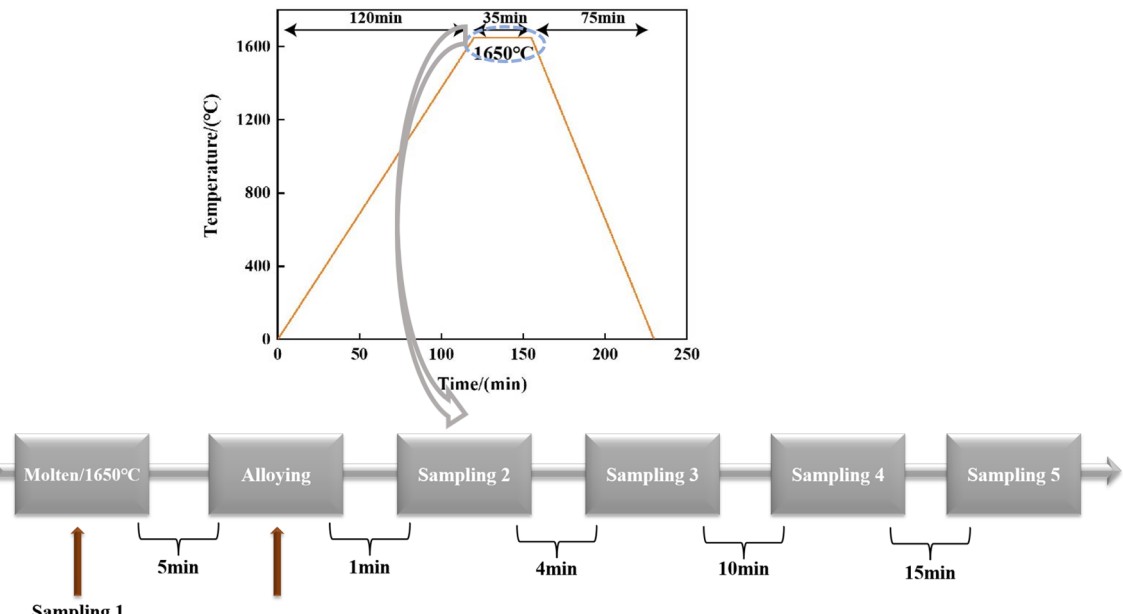

**Figure 2.** Heating cycle used in this study.

As this experiment focused on typical aluminum-deoxidized steel-bearing steel, the silicon and manganese content varied within the bearing steel composition range. According to the GB/T1825-2016 high-carbon chromium-bearing steel national standard, several groups of different Mn/Si ratios were established [28]. The scheme is shown in Table 2.

**Table 2.** Experimental plan.

| Case | Mn/wt % | Si/wt % | Mn/Si | Initial T.O/$10^{-6}$ |
|------|---------|---------|-------|-----------------------|
| 1 | 0 | 0.2 | Pure Si | 100 |
| 2 | 0 | 0.2 | Pure Si | 300 |
| 3 | 0 | 0.2 | Pure Si | 800 |
| 4 | 0.25 | 0.35 | 0.71 | 300 |
| 5 | 0.34 | 0.2 | 1.71 | 300 |
| 6 | 0.45 | 0.15 | 3.0 | 300 |

T.O means initial total oxygen content.

## 2.2. Sample Characterization

The experimental samples were processed using wire-cutting equipment to obtain samples of different shapes and sizes according to experimental needs (Figure 1b). The Φ5 mm × 8 mm round bar samples obtained by cutting were analyzed using an oxygen–nitrogen–hydrogen analyzer (LECO TCH-600, LECO, St. Joseph, MI, USA). The Φ20 mm × 3 mm slices obtained by cutting the steel ingot were polished to a bright surface, and composition analysis was conducted using a direct reading spectrometer (ARL-8860, Thermo Fisher Scientific, Waltham, MA, USA). The measured Si and Mn content in the steel is shown in Table 3.

For samples with dimensions of 10 mm × 10 mm × 10 mm obtained by cutting, they were grounded and polished by using sandpaper and polishing cloths. The size, distribution, and number of inclusions larger than 1 μm were quantified by using an automated scanning electron microscope (ASPEX, Reston, VA, USA). Inclusions' composition and 2D morphology were analyzed by a scanning electron microscope (SEM, Phenom-ProX, Utrecht, The Netherlands) equipped with an energy dispersive X-ray spectrometer (EDS).

In addition, the three-dimensional morphology of inclusions was observed by small-sample electrolysis and suction filtration (Figure 3). In this process, the samples were used as the anode, and a cylindrical stainless-steel cylinder served as the cathode, fixed in solution A (78% methanol, 10% glycerin, 10% triethanolamine, and 2% tetramethylammonium chloride, and the total volume was 200 mL). The electric current was controlled between 50 mA and 70 mA with a voltage of 7 V by using a regulated DC power supply (Model DH1720A-1), while the temperature of the electrolyte was kept at room temperature. After 6 h of electrolysis, the anode was cleaned with alcohol several times and dried before analysis via SEM/EDS analysis.

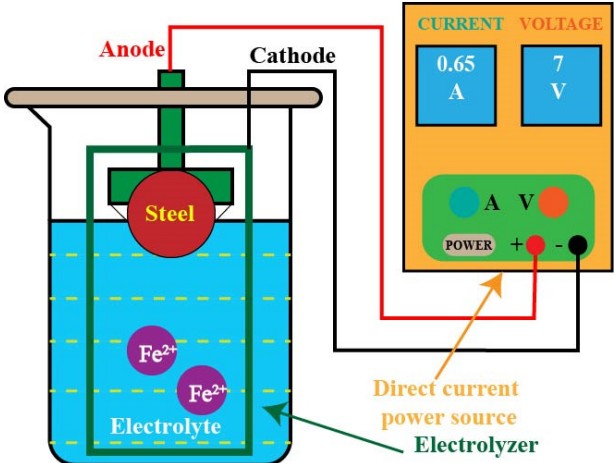

**Figure 3.** Non-aqueous solution electrolysis method.

**Table 3.** Composition of steel, wt %.

| Sample | [C] | [Si] | [Mn] |
|--------|-----|------|------|
| Case 2 | 0.013 | 0.19 | 0.009 |
| Case 4 | 0.005 | 0.33 | 0.23 |
| Case 5 | 0.005 | 0.22 | 0.34 |
| Case 6 | 0.012 | 0.12 | 0.38 |

## 3. Results and Discussion

### 3.1. Oxygen Content Changes

#### 3.1.1. Effect of Different Initial Oxygen Contents on Deoxidation

The oxygen content of the experimental samples was influenced by factors such as the endpoint temperature, the endpoint carbon content, and technology. Therefore, three experiments with different initial oxygen contents were designed to determine the deoxidizing ability of [Si] addition. Figure 4a shows the change in content of oxygen in molten steel with different initial oxygen contents using low-carbon ferrosilicon deoxidation. As observed in Figure 4a in Case 3, after adding the ferrosilicon alloy for 1 min, a vigorous reaction occurred between the ferrosilicon and the oxygen in the molten steel, resulting in a substantial drop in oxygen content from 800 ppm to 307 ppm. Afterward, the change in oxygen content gradually flattened. After the molten steel reacted with ferrosilicon for 5 min, the oxygen content dropped from 307 ppm to 160 ppm, and after 15 min, it decreased from 160 ppm to 94 ppm. Following 30 min of reaction, the oxygen content stabilized at 60 ppm. The oxygen content curve in Case 2 exhibited a more consistent change pattern. After 5 min of the reaction, the oxygen content dropped from 307 ppm to 160 ppm. The trend in oxygen content in Case 2 resembled that of Case 3, and ultimately, after 30 min of the reaction, the oxygen content in Case 2 stabilized at 35 ppm. The oxygen content in the molten steel of Case 1 was also stable at 35 ppm in the end. The higher the initial oxygen content concentration of the molten steel, the more significant the decrease in the total oxygen content fraction after adding ferrosilicon for 1 min. As the oxygen content in molten steel increased, the deoxidation rate of the Si gradually increased. This is because as the [O] content in molten steel increases, the [Si] element in molten steel is more likely to capture the [O] element in molten steel after adding ferrosilicon. Therefore, the case with a high initial oxygen content has a high deoxidation rate. It was observed that Si deoxidation controlled the oxygen content in molten steel to 43 ppm.

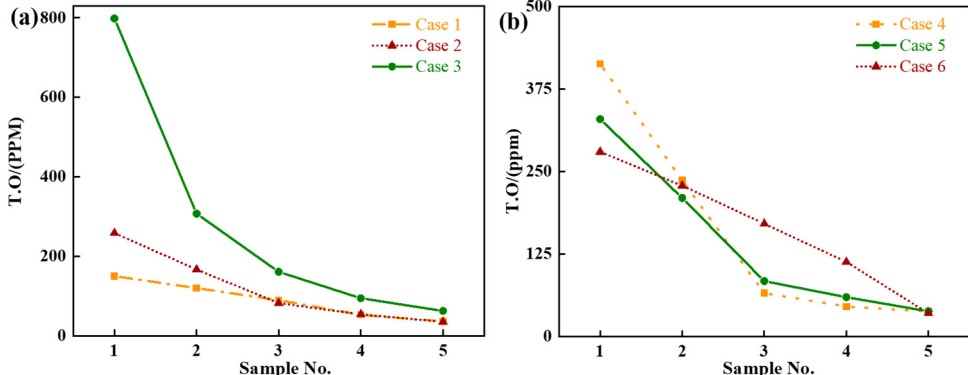

**Figure 4.** Changes in oxygen content in molten steel: (**a**) different initial oxygen; (**b**) different Mn–Si ratio.

#### 3.1.2. Effect of Different Mn–Si Ratios on Deoxidation

When silicon was used for deoxidation, the deoxidation product $SiO_2$ exhibited a melting point as high as 1710 °C and existed in a solid state in the molten steel. Furthermore, when Si–Mn was used for deoxidation, a low-melting-point compound ($SiO_2$–MnO) was

formed, which was conducive to aggregation, growth, and floating removal. The compound reduced $SiO_2$ activity and improved the deoxidation ability of Si. In this study, three experiments with different Mn–Si ratios were established. Figure 4b displays the change in total oxygen content in molten steel with different Mn–Si ratios with initial oxygen content of 300 ppm. After the addition of the Si–Mn alloy into the molten steel for 5 min, the oxygen content of Case 4 dropped significantly from 400 ppm to 66 ppm. Afterward, the oxygen content change curve became flattened. After 30 min of the reaction, the oxygen content of Case 4 stabilized at 35 ppm. Similarly, in Case 5, the oxygen content dropped significantly from 330 ppm to 84 ppm after 5 min of the reaction, with a subsequent gradual decrease in the oxygen content curve. After 30 min, the oxygen content in Case 5 stabilized at 36 ppm. The oxygen content change curve in Case 6 exhibited a more consistent pattern, with a gradual reduction from 280 ppm to 35 ppm. Notably, the lower the Mn–Si ratio, the more pronounced the decrease in the total O mass fraction within 5 min. This was because [Si] exhibited a stronger binding ability with [O] than [Mn]. As the Si content decreased, the deoxidation rate decreased. Ultimately, the use of Si–Mn alloy deoxidation effectively controlled the oxygen content in molten steel to 35 ppm.

When silicon–manganese was used for deoxygenation within the first 15 min of the reaction, Case 4 had the highest deoxygenation efficiency, Case 5 had the second highest deoxygenation efficiency, and Case 6 had the lowest deoxygenation efficiency. This is because [Si] is more reducible than [Mn]. Therefore, when the Mn–Si ratio is low, [Si] as the main element of deoxidation is high and the reduction rate is faster. However, as the Mn–Si ratio increases, the [Si] content gradually decreases, and the deoxidation rate also slows down.

### 3.1.3. Thermodynamic Calculations

The reaction formula involving [Si], [Mn], and [O] is shown in Formulas (1) and (3) [29,30]. When Si and Mn are simultaneously deoxidized, the reaction in Formula (5) is followed. The expressions of Si and Mn deoxidation equilibrium constants are Formulas (2) and (4). [Si], [Mn], and [O] in molten steel conformed to Henry's law. However, the activities of deoxidized products of $SiO_2$ and MnO varied with the ratio of Mn–Si in molten steel. The Mn–Si ratios in the experiments were 0.71, 1.55, and 3.17. To begin, the $SiO_2$ activity was calculated for these three sets of Mn/Si ratios using Factsage 7.2 software. According to Formulas (6)–(8), the change in [O] content in the molten steel equilibrium with [Si] was obtained (Figure 5). Additionally, it was observed from Figure 5 that when the Mn–Si ratio was 3.17, the mass fraction of [O] was 35 ppm, corresponding to the experimental results. During the actual production process, the content of the ferrosilicon added was excessive, such that in the case of the three Mn–Si ratios, the content of [O] in molten steel was stabilized at 35 ppm. When the Mn–Si ratio was 0.71, the content of [O] was 90 ppm. When the Mn–Si ratio was 1.55, the oxygen content was 70 ppm. The value obtained from the experimental results deviated from the calculated value. This was attributable to the crucible material (MgO) used in this experiment, which reacted with $SiO_2$ at high temperatures to form $MgSiO_3$. It influenced the reaction equilibrium of Formula (1) to shift to the right. Deoxidation continued until equilibrium was reached, resulting in the experimental oxygen content being lower than the calculated oxygen content.

$$[Si] + 2[O] = (SiO_2) \quad \Delta G^{\theta} = -581{,}900 + 221.8T \tag{1}$$

$$lgk_{Si} = \frac{a_{SiO_2}}{a_{[Si]} \cdot a_{[O]}^2} = \frac{30{,}391}{T} - 11.58 \tag{2}$$

$$[Mn] + [O] = (MnO) \quad \Delta G^{\theta} = -24{,}430 + 108.8T \tag{3}$$

$$lgk_{Mn} = \frac{a_{MnO}}{a_{[Mn]} \cdot a_{[O]}} = \frac{12{,}759}{T} - 5.68 \tag{4}$$

$$[Si] + 2(MnO) = (SiO_2) + 2[Mn] \quad \Delta G^{\theta} = -93{,}300 + 4.2T \tag{5}$$

$$w_{[O]} = \frac{\sqrt{\frac{a_{SiO_2}}{a_{[Si]}\cdot\left(\frac{30,391}{T}-11.58\right)}}}{f_{[O]}} = \frac{\sqrt{\frac{a_{SiO_2}}{f_{[Si]}\cdot w_{[Si]}\left(\frac{30,391}{T}-11.58\right)}}}{f_{[O]}} \qquad (6)$$

$$lgf[O] = e_O^{Si}\cdot\%[Si] + e_O^{Mn}\cdot\%[Mn] \qquad (7)$$

$$lgf[Si] = e_{Si}^{O}\cdot\%[O] + e_{Si}^{Mn}\cdot\%[Mn] \qquad (8)$$

$k_{Si}$—equilibrium constant of Si; $a_{SiO_2}$—activity of $SiO_2$; $a_{[Si]}$—activity of Si; $a_{[O]}$—activity of O; $k_{Mn}$—equilibrium constant of Mn; $a_{MnO}$—activity of MnO; $a_{[Mn]}$—activity of Mn; $w_{[O]}$—mass fraction of O; $f_{[O]}$—activity coefficient of O; $e_O^{Si}$—activity interaction coefficient of Si to O; $e_O^{Mn}$—activity interaction coefficient of Mn to O; $f_{[Si]}$—activity coefficient of Si; $e_{Si}^{O}$—activity interaction coefficient of O to Si; $e_{Si}^{Mn}$—activity interaction coefficient of Mn to Si.

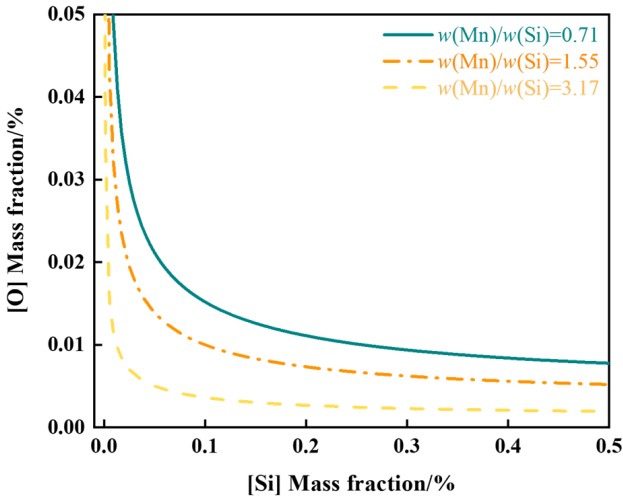

**Figure 5.** Variation in [O] mass content in [Si] equilibrium molten steel.

*3.2. Inclusion Changes*

3.2.1. Number and Size of Inclusions

The various types of inclusions in the samples of Si-deoxidized and Si–Mn-deoxidized molten steel were analyzed by using automatic scanning electron microscopy (ASPEX). We performed automatic scanning over an area of 5 mm × 5 mm of the metallographic sample. Inclusions with Al, Mg, Ca, Si > 0 and Al + Mg + Ca + Si > 1% were screened and categorized. Then, the number and size of inclusions that satisfied the above requirements were determined. The results are shown in Figure 6. It was observed that the size of inclusions changed significantly as the initial oxygen content increased. The inclusion size of Case 1 stabilized within the range of 1–2.5 μm. The inclusion size of Case 2 stabilized at 4 μm. However, the inclusion size of Case 3 was as high as 6 μm. In addition, the density of $SiO_2$ increased as the initial oxygen content of the steel increased. There were very few $Al_2O_3$, Mg–Al spinel, $MgO$-$SiO_2$, $CaO$-$SiO_2$, and $CaO$-$Al_2O_3$-$SiO_2$ inclusions in another case. Although the amount of [Al] and [Mg] in electrolytic iron is very small, some elements in the crucible refractory entered the molten steel during heating. As the initial oxygen content increased, the density of $SiO_2$ increased from 2.6·mm$^{-2}$ to 43.07·mm$^{-2}$. This is because the high oxygen content increased the number of inclusions.

It was observed from Figure 6c,d that with the increase in the Mn–Si ratio, for Case 4, the inclusion size peak was located at 6 μm. The crucible delivered Ca and Mg and a certain amount of Al to the molten steel. As a result, a certain number of $Al_2O_3$, CaO, $MgO$-$SiO_2$, $CaO$-$SiO_2$, and $Al_2O_3$-$SiO_2$ inclusions were generated in molten steel. However, most were still $SiO_2$ in this case. When Si and Mn were deoxidized at the same time, Mn gradually reacted with $SiO_2$ to form liquid $MnO$–$SiO_2$. Thus, the density of $SiO_2$ gradually decreased

with the increased Mn–Si ratio. Despite the presence of $Al_2O_3$ inclusion types, the inclusion density of Case 6 was still the lowest. The [Si] content of the alloy gradually decreased as the Mn-Si ratio was elevated. So, the $SiO_2$ inclusions were also gradually reduced. At the same time, the addition of [Mn] also generated $MnO$–$SiO_2$ (low-melting-point composite inclusions), which promoted the removal of $SiO_2$ inclusions by aggregation.

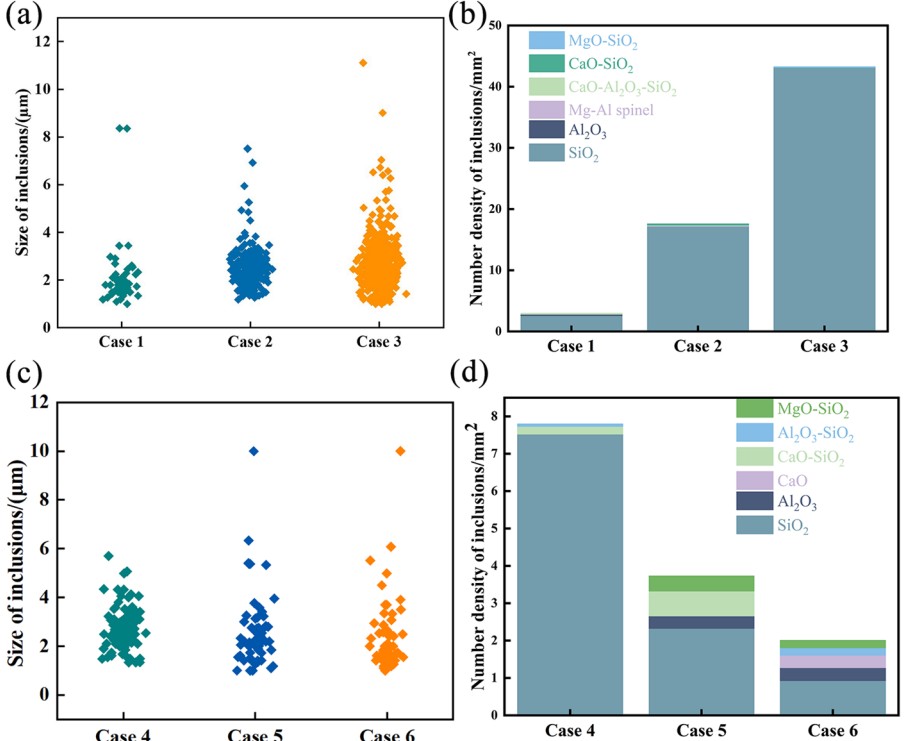

**Figure 6.** Size distribution and numerical density of typical inclusions in samples with different cases. (**a**,**b**) Different initial oxygen levels; (**c**,**d**) different Mn–Si ratios.

The average composition (mass fraction) of $SiO_2$ inclusions in steel was calculated using Factsage thermodynamic software under different Mn–Si ratios for comparison. The FToxide database was selected, and the calculation results are shown in Figure 7. With an increased Mn–Si ratio, the mass fraction of $SiO_2$ inclusions gradually decreased.

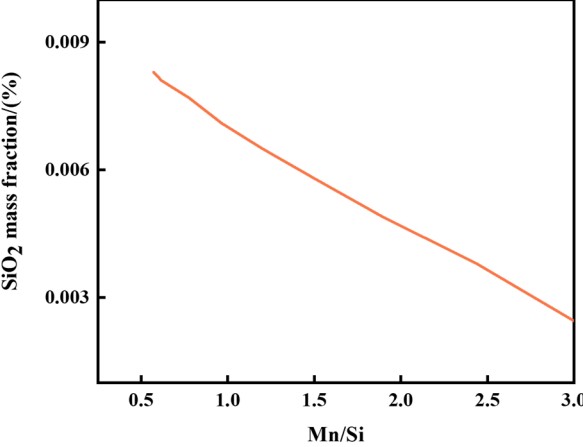

**Figure 7.** Thermodynamic calculation of inclusion mass fraction in different Mn–Si ratios.

### 3.2.2. Inclusion Morphology and Composition

The morphology of inclusions was observed using a scanning electron microscope (SEM-EDS) equipped with an energy dispersive spectrometer during the sampling of Si-deoxidized steel (Figure 8a–c). Because it was primarily deoxidized using [Si], the main inclusions in the steel were silicates. Furthermore, due to the different initial oxygen contents, silicates ranging from 1 μm to 6 μm are listed below. The inclusion size of Case 1 was mainly concentrated at 1 μm. The inclusion size of Case 2 was mainly concentrated at 3 μm. The inclusion size of Case 3 was mainly concentrated at 6 μm. It was observed from the figure that silicates and the composite inclusions containing Si were nearly spherical.

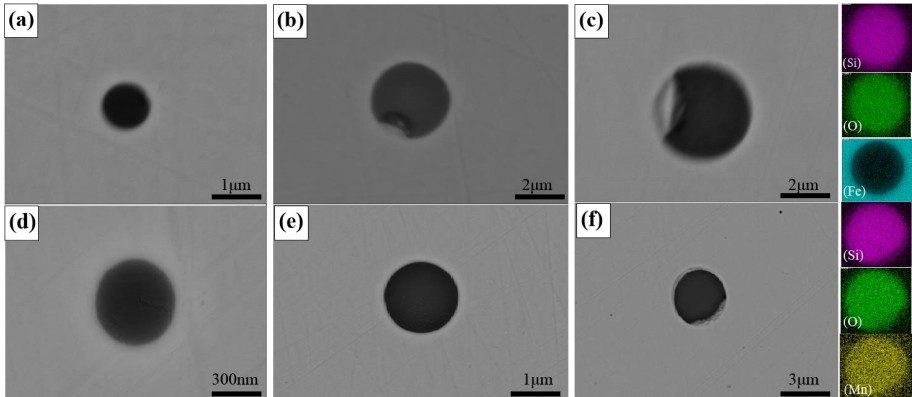

**Figure 8.** Two-dimensional morphologies of inclusions in steel: (**a**) Case 1; (**b**) Case 2; (**c**) Case 3; (**d**) Case 4; (**e**) Case 5; and (**f**) Case 6.

In this study, a small-sample electrolysis and suction filtration experiment on Case 1 was conducted, and the inclusions observed under SEM-EDS, as shown in Figure 9a–c, still exhibited irregular strip- and cube-like shapes. Furthermore, as shown in Figure 8d–f, during the silicon–manganese-deoxidized molten steel process, the main inclusions in the steel remained silicates, and [Mn] primarily enhanced silicon deoxidation. However, inclusions containing [Mn] were less frequent. A small-sample electrolysis and suction filtration experiment conducted on Case 6 revealed that the inclusions filtered were composite inclusions of silicates and $Al_2O_3$. The 3D morphology of these inclusions was irregular and cube-like (Figure 9d–f).

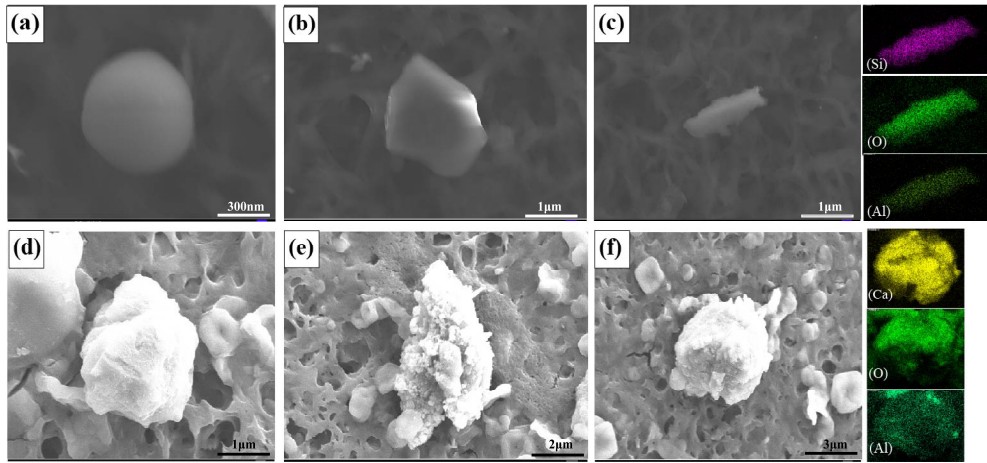

**Figure 9.** Three-dimensional morphologies of inclusions in steel: (**a**–**c**) Case 1; (**d**–**f**) Case 6.

### 3.2.3. Inclusion Formation Mechanism Analysis

Comparing the composition and size of inclusions in Figure 10, it was inferred that the formation mechanism of Si–Mn weakly deoxidized inclusions during induction heating

and followed the schematic diagram shown in Figure 11. Initially, during the smelting process, [O] in molten steel reacted with [Si] to form $SiO_2$. After that, the MnO inclusions were covered to the outer layer of the $SiO_2$ inclusions. Afterward, MnO gradually reacted with $SiO_2$ to form liquid $MnO-SiO_2$. The $MnO-SiO_2$ substituted the previous $SiO_2$ and the inclusions were spherical. As the reaction proceeded, the amount of $MnO-SiO_2$ gradually increased. The composite inclusions of $MnO-SiO_2$ collided with each other to form composite inclusions of $MnO-SiO_2$. These inclusions remained spherical because of the liquid nature of $MnO-SiO_2$, eventually growing and stabilizing.

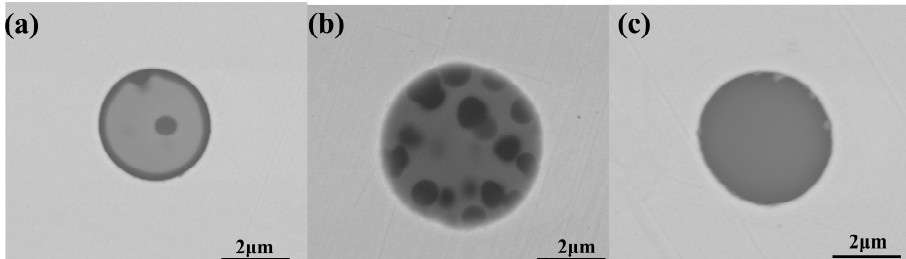

**Figure 10.** Inclusion morphology at different stages in Case 6: (**a**) 1 min, (**b**) 15 min, and (**c**) 30 min.

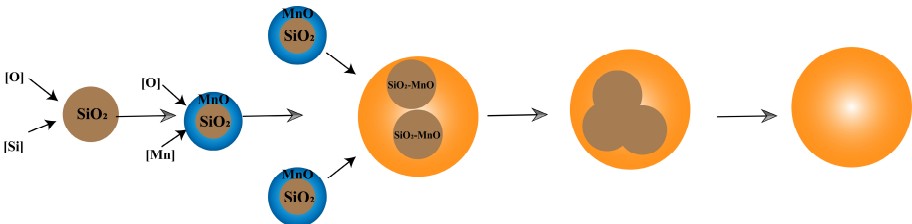

**Figure 11.** The formation process of inclusions during Si–Mn deoxidation.

## 4. Conclusions

In this study, the effect of Mn and Si content on the characteristics of inclusions in steel was investigated through laboratory thermal experiments and thermodynamic calculations. The main conclusions were obtained as follows:

(1)  Silicon–manganese deoxidation exhibited weaker deoxidation abilities compared with aluminum deoxidation. However, the Si deoxidation process reduced the oxygen content in molten steel to 43 ppm, while the silicomanganese deoxidation process's final oxygen content was 35 ppm. The deoxidation rate of the silicon–manganese process was lower compared with the silicon deoxidation process.

(2)  Silicon–manganese deoxidation significantly reduced the content of B-type inclusions in steel and generated silicate inclusions. The size and density of inclusions peaked at a Mn–Si ratio of 0.71. Afterward, the size and density of inclusions in the steel decreased with an increased Mn–Si ratio.

(3)  As the reaction progressed, inclusions in the molten steel initially appeared as spherical $SiO_2$-type inclusions of 1 μm. Gradually, [Mn] combined with liquid $SiO_2$ to form $MnO-SiO_2$ composite liquid inclusions of 1 μm. Finally, $MnO-SiO_2$ aggregated and reached a stable state.

**Author Contributions:** Writing—original draft preparation: T.S.; writing—review and editing: Z.W. and Y.B.; funding acquisition, Y.B.; methodology, C.G.; resources, Z.Z.; validation, Y.B., C.G. and Z.Z. All authors have read and agreed to the published version of the manuscript.

**Funding:** This research was funded by National Natural Science Foundation of China (52174297).

**Data Availability Statement:** Data are contained within the article.

**Acknowledgments:** The authors wish to express their gratitude to the foundation for providing financial support.

**Conflicts of Interest:** Ze-Feng Zhang was employed by the company HBIS Materials Technology Research Institute. The remaining authors declare that the research was conducted in the absence of any commercial or financial relationships that could be construed as potential conflicts of interest.

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
