# Peer review of "Effect of Silicon–Manganese Deoxidation on Oxygen Content and Inclusions in Molten Steel"

_processes, doi:10.3390/pr12040767_

Round 1

Reviewer 1 Report

Comments and Suggestions for Authors

The manuscript is well written. Here are few comments twhic can help improve the quality of the manuscript

The introction and literature review needs improvement

The outcomes of the literature survey should be highlighted clearly at the end of the introduction section, indicating the lacuna and the scope and nature of the work carried.

The authors conducted only six experiments, Using these limited experiments it is possible to draw clear conclusions?

At certain places, there is a space between the entity and its unit (6 h), and in some places, the space is not provided (200ml). Uniformity can be maintained.

Authors stste that high initial oxygen content has a high deoxidation efficiency. Any particular reason? The reason should be explained.

When Si–Mn was used for deoxidation, a low-melting point compound (SiO2-MnO)was formed. How was the formation verified?

Comments on the Quality of English Language

Needs minor improvement

Author Response

Dear reviewer:

Thanks for your valuable comment. I've attached my reply below.

Reviewer 2 Report

Comments and Suggestions for Authors

Dear Authors

I read your paper with interest.

I believe that the following changes or clarifications should be made to the text.

1. Why was bearing steel chosen for testing?

2. How many experiments were performed?

3. Was the steel cast into a mold or left in a crucible to solidify?

4. What was the pressure in the furnace space?

5. Specific requirements of different steel grades cannot affect the oxygen content. Chemical composition and technology may have an impact. (chapter 3.1.1)

6. What statistical analysis was performed? It should be described in the research methodology. No statistical analysis results. How much and which data was analyzed? (chapter 3.2.1)

7. The fluidity of steel has not been tested, so the conclusion is indirect. Conclusion 2 should therefore be reworded.

Author Response

(The authors gave the same response as above.)

Round 2

Reviewer 2 Report

Comments and Suggestions for Authors

Thank you for the changes.

There is still no information about the scope and results of the statistical analysis, which the authors write about in line 236.

Please indicate which data were subjected to which statistical tests and what results were obtained.

Author Response

Dear reviewer:

I apologize for not fully understanding your last question.I answered the questions you asked this time.

Round 3

Reviewer 2 Report

Comments and Suggestions for Authors

Dear Authors

According to the authors' explanations, the data were not subject to statistical analysis. Please remove the word "statistically" from line 236.

Author Response

Dear reviewer:

Thanks for your valuable comment. I have already deleted the word.